# Carboxy-Methylation of the Catalytic Subunit of Protein Phosphatase 2A (PP2Ac) Integrates Methionine Availability with Methionine Addicted Cancer Cell Proliferation

**DOI:** 10.3390/biom15091210

**Published:** 2025-08-22

**Authors:** Anna Andronicos, Kiku C. Yoneda, Da-Wei Lin, Fiona V. Law, Hosung Bae, Ali Basirattalab, Nicholas A. Graham, Cholsoon Jang, Peter Kaiser

**Affiliations:** 1Department of Biological Chemistry, University of California, Irvine, CA 92697, USA; ageorgi1@uci.edu (A.A.); kyoneda@uci.edu (K.C.Y.); ddlin@hs.uci.edu (D.-W.L.); lawf@uci.edu (F.V.L.); hosungb@hs.uci.edu (H.B.); choljang@uci.edu (C.J.); 2Mork Family Department of Chemical Engineering and Materials Science, University of Southern California, Los Angeles, CA 90089, USAnagraham@usc.edu (N.A.G.); 3Norris Comprehensive Cancer Center, University of Southern California, Los Angeles, CA 90089, USA; 4Department of Chemistry, University of California, Irvine, CA 92697, USA

**Keywords:** methionine dependence of cancer, methionine restriction, Hoffman effect, protein phosphatase 2A, methylation

## Abstract

Cancer cells exhibit a well-documented, yet poorly understood, dependence on exogenous methionine, despite retaining the capacity to convert homocysteine to methionine. In contrast, non-tumorigenic cells can proliferate when methionine is replaced by homocysteine. To investigate the mechanistic basis of this methionine dependence, we examined how methionine metabolism impacts cancer cell proliferation. We identified carboxy-methylation of the catalytic subunit of Protein Phosphatase 2A (PP2A) as a critical node linking methionine availability to proliferation. PP2A methylation was found to be highly sensitive to intracellular S-adenosylmethionine (SAM) levels, with reduced methylation correlating with impaired proliferation under methionine restriction. Overexpression of Protein Phosphatase Methylesterase-1 (PME-1), which demethylates PP2A, or expression of a Leu309-deleted PP2A mutant that mimics the demethylated form, was sufficient to reduce proliferation even in methionine-independent cells. These findings support a model in which methionine limitation lowers SAM availability, thereby decreasing PP2A methylation and impairing cell proliferation. Our study reveals a mechanistic link between methionine metabolism and cell proliferation and suggests that PP2A methylation plays a key role in the unique methionine dependence of cancer cells.

## 1. Introduction

Since the 1900s, dietary restriction has been linked to longer lifespans and better prognoses for many diseases [1], although the mechanisms remain poorly understood. Restriction of the essential amino acid methionine has especially been linked with improved metabolic health [2] and extended lifespan in model organisms, including rodents [3,4,5]. In addition to extending longevity, methionine-restricted (MR) diets have shown positive effects in cancer models. As early as 1959, MR diets were reported to significantly reduce tumor growth in tumor-bearing rats [6]. Since then, these anti-tumor effects have been replicated in numerous animal models [7,8,9]. The tumor-suppressive activity of MR diets is thought to stem from a cancer cell–intrinsic reliance on exogenous methionine, a phenomenon termed methionine dependence of cancer, methionine addiction of cancer, or the Hoffman Effect [10,11]. This phenomenon is defined by a cancer-specific metabolic addiction showing that most cancer cells, but not normal cells, are dependent on exogenous methionine and are unable to proliferate when methionine is replaced with its precursor, homocysteine [12,13,14]. This methionine addiction has been shown to be tightly correlated with malignancy [15]. Using murine leukemia cells, this phenomenon was first reported in 1973 [13] and has since then been further developed in a multitude of studies [11,12,14,16,17,18]. While it has been shown that cancer cells are able to synthesize methionine from homocysteine [11], it is not fully understood why they cannot proliferate in these conditions. One proposed mechanism underlying methionine dependence in cancer is an increased rate of transmethylation reactions that exceeds the capacity for metabolic regeneration [19,20]. Consistent with this idea, isotope tracing and metabolite add-back studies have shown that culturing cancer cells in homocysteine reduces S-adenosylmethionine (SAM) synthesis, thereby limiting the transmethylation capacity required to support rapid proliferation [20,21,22,23]. Additionally, impaired methionine synthase activity due to limited assembly of the active holoenzyme with its vitamin B12-derived cofactor, methylcobalamin, has been suggested as a contributing factor in certain cancer cell systems [24,25]. However, other studies have reported that cancer cells require functional methionine synthase holoenzyme for survival due to its essential role in folate generation [26,27].

Transmethylation and SAM production are tightly coupled to methionine metabolism, as methionine not only serves as a proteogenic amino acid but also reacts with ATP to form SAM, the primary methyl donor in cellular processes. As such, SAM is responsible for most methylation events, including methylation of DNA, RNA, proteins, and lipids. After loss of the methyl group, the metabolite S-adenosyl homocysteine (SAH) is hydrolyzed into adenosine and homocysteine, which then becomes the precursor for methionine in the methionine cycle [10] (Figure 1A). In methionine-limiting conditions, the SAM/SAH ratio, a measure of cellular methylation potential, significantly decreases in tumorigenic cells [7,23,24]. A possible explanation for decreased SAM levels observed during growth in homocysteine, specifically in tumor cells, is that homocysteine metabolism is redirected towards the transsulfuration pathway and glutathione synthesis, as indicated by stable isotope tracing [23]. This metabolic shift likely occurs in response to increased oxidative stress often experienced by cancer cells. When combined with the elevated transmethylation demands associated with cancer cell proliferation [19,20], these metabolic constraints may trigger a protective cell cycle arrest that progresses to induced cell death if stress persists [18,28,29]. It has been suggested that monitoring cellular methylation potential and halting proliferation in response to SAM depletion constitutes a crucial evolutionarily conserved cell cycle checkpoint that safeguards epigenetic and cellular integrity [12,18,29,30]. While several pathways sensing methionine availability and its related metabolites have been identified, the mechanisms by which the cell proliferation machinery detects decreases in methionine or SAM levels and communicates with the cell cycle and survival machinery remain poorly understood [30].

Extensive research in yeast has uncovered multiple pathways that sense methionine metabolism status. For instance, translation capacity is regulated by the availability of sulfur-containing amino acids through tRNA thiolation [31]. Additionally, methionine levels modulate the expression of metabolic enzymes involved in methionine synthesis and cell cycle regulation by influencing the interaction of transcription factors with the ubiquitin ligase SCFMet30 [32,33,34,35]. Furthermore, autophagy is linked to methionine metabolism through the carboxymethylation of Protein Phosphatase 2A (PP2A), highlighting the intricate regulatory network governing methionine sensing and utilization [36].

In mammalian cells, several mechanisms have been implicated in sensing methionine and SAM availability. The PRMT1-SAMTOR axis serves as a sensor of SAM levels, linking methionine metabolism to mTORC1 signaling [37,38]. Additionally, splicing regulation via SmD1 methylation highlights another layer of methionine-dependent control over gene expression [39]. Despite these mechanisms, neither SAMTOR signaling nor SmD1 methylation appears to rapidly and directly connect methionine metabolism to cell proliferation [29,39]. In this report, we propose that PP2A methylation acts as a critical mediator of methionine addiction in cancer cells and represents a major mechanism linking methionine metabolism to cell proliferation. PP2A is a heterotrimeric complex that consists of the structural subunit A, the catalytic subunit C, and one of several regulatory subunits B, which determine substrate specificity [40,41,42]. PP2Ac is the subunit that can be carboxymethylated on its terminal amino acid, Leu309, by the PP2A-specific methyltransferase LCMT-1 [43,44]. The methylation status of PP2Ac has been linked with changes in B-subunit binding [44,45].

Given the role of PP2A as a tumor suppressor that regulates key signaling pathways to control cell cycle progression and survival, its methylation status may serve as a functional readout of methionine availability, providing a direct and dynamic connection between metabolic state and cancer cell proliferation.

## 2. Materials and Methods

### 2.1. Cell Lines and Growth Conditions

The triple-negative breast cancer cell line MDA-MB-468 and MB-468res-R8 [23] and HA-PP2Ac HEK293T cells [46] were used. The HEK293Tres-R1 cells were generated as described in Hoffman et al. [47] and Borrego et al. [48]. All cell lines used were purchased from ATCC: MDA-MB-468-HTB-132-ATCC, 293T-CRL-3216-ATCC, MDA-MB-231-HTB-26-ATCC, PANC1-CRL-1469-ATCC, BxPC3-CRL-1687-ATCC. Cells were maintained in DMEM (Corning, Corning, NY, USA, 10-017-CV) supplemented with 10% FBS (Gibco, Grand Island, NY, USA, A52567-01) and 1% of antibiotic (Corning, Corning, NY, USA, 30-004-CI).

For all metabolic shift experiments, cells were grown in DMEM media until appropriate confluence was reached, cells were then washed with PBS and treated with methionine-containing, methionine-free, or homocysteine media. All limiting media is prepared with DMEM (high glucose, no glutamine, no methionine, no cystine—Gibco, 21013-024) supplemented with 10% dialyzed FBS (Omega Scientific, Tarzana, CA, USA, FB-03) and 1% of antibiotic, 1.5 μM cyanocobalamin (vitamin B12, Biotech BP862-1), 4 mM L-glutamine (Fisher Scientific, Waltham, MA, USA, O2956100), 100 μM L-cysteine (Fisher Scientific, BP376100), and 100 μM L-methionine (Sigma-Aldrich, St. Louis, MO, USA, M5308). In the case of homocysteine media, 100 μM L-homocysteine (Biosynth, Staad, Switzerland, FH52361) was added in the absence of methionine. Previous reports from our lab using MDA-MB468 cells have also used 370 μM DL-homocysteine (Sigma-Aldrich, St. Louis, MO, USA, H4628) due to limited availability of high-quality L-homocysteine [18,23,48,49]. We did not observe significant changes in growth behavior between supplementation with 100 µM L-Hcy and 370 µM DL-Hcy Appendix A.

### 2.2. PME-1 Overexpression Cell Lines

Vector PLX304-PME1-V5 (GE Dharmacon, Lafayette, CO, USA) containing the human PME-1 sequence with V5 tag was transfected into HEK-293T cells plated to 70% confluence with lentiviral packaging vector psPAX2 (Addgene, Watertown, MA, USA, #12260) and envelope vector PMD2.G (Addgene, Watertown, MA, USA, #12259) using BioT transfection reagent (Bioland, Belgrade, Serbia, #B01-01) and serum-free DMEM media. Control lentivirus was generated at the same time using all identical transfection components except PLX304-PME1-V5 vector. Lentiviral supernatant was collected at 24, 48 and 72 h post transfection. Cell lines HEK293T, R1, MDA-MB468, R8, MDA-MB231, BxPC3, and PANC1 were plated to 70–80% confluence in 6-well plates in serum-free DMEM media and infected with a combination of 0.5 mL 24-h post transfection and 0.5 mL 48-h post transfection viral supernatant, and polybrene at [1 mg/mL] concentration. After 48 h, cells were split and selected with varying concentrations of Blasticidin [2–10 µg/mL] until complete death of cells infected with control viral supernatant. Expression of PME-1 was verified via western blotting with PME-1 antibody (Millipore, Burlington, MA, USA, Anti-PME1, clone 8A6-F8) at 1:2000 dilution.

### 2.3. ∆Leu309 Cell Lines

The vector PQCXIH-mycPP2Ac-∆Leu309 containing the human PP2Ac coding sequence with a myc tag and a terminal amino acid, Leucine 309, deletion was used. R8 and R1 cells were plated and at 70% confluence, cells were transfected according to the manufacturer’s instructions using 10 µg of plasmid and 20 µL Lipofectamine 2000 (Invitrogen, Carlsbad, CA, USA, 11667-027) in 150 µL OPTI-MEM media. Media was changed after 24 h, and after 48 h, cells were selected with 8 µL Hygromycin B (Enzo, Lausanne, Switzerland, ALX-380-059-UMO1) for 7 days.

### 2.4. Growth Curves, Viability Assay, IC_50_ Determination

Cell lines were grown overnight in complete DMEM media and then transferred to either a 96-well plate (2000 cells seeded) or a 24-well plate (25,000 cells seeded) with their respective media (methionine media/−Met+Hcy media). The Sartorius IncuCyte S3 Live Cell Analysis System was used to capture images and analyze data.

Cell viability assays to determine IC50 values for doxorubicin were performed using CellTiter-Glo Luminescent Cell Viability Assay (Promega, Wisconsin, MD, USA) according to the manufacturer’s instructions. Cells were seeded in a 96-well plate at a density of 2000 cells/well. After 24 h, cells were treated with either vehicle (water) or varying concentrations of Doxorubicin (DOX). For MB468s and R8s, the concentrations ranged from 5 µM to 0.0003 µM, while for the HEK293T and R1s, the range was 20 µM to 0.001 µM. IC_50_ values for the compounds were calculated using GraphPad Prism version 10.

### 2.5. Protein Analysis

Whole cell lysates were prepared in Triton X-100 buffer (50 mM HEPES, pH 7.5, 0.2% Triton X-100, 200 mM NaCl, 10% glycerol, 1 mM dithiothreitol, 10 mM Na-pyrophosphate, 5 mM EDTA, 5 mM EGTA, 50 mM NaF, 0.1 mM orthovanadate, 1 mM phenylmethylsulfonyl fluoride [PMSF], and 1 mg/mL of leupeptin and pepstatin). Cells were lysed and lysates were homogenized using a disposable 27G needle. Lysates were cleared by centrifugation (15 min, 16,000× *g* at 4 °C). For Western blot analyses, proteins were separated by SDS-PAGE and transferred to a polyvinylidene difluoride (PVDF) membrane. Proteins were detected using the following primary antibodies: methyl PP2Ac (Santa Cruz, CA, USA, sc-81603, demethylated PP2Ac (Santa Cruz, CA, USA, sc-13601), total PP2Ac (Proteintech, Rosemont, IL, USA, 13482-1-AP), PARP (Cell Signaling, Danvers, MA, USA, 9542S) Cleaved Caspase-3 (Cell Signaling, 9661S), pS6 (Cell Signaling, 2317S), S6 (Cell Signaling, 4838S), LC3 A/B (Cell Signaling, 4108S), PME-1 (Millipore, MABC1183), p4EBP1 (Cell Signaling, 9459S), 4EBP1 (Cell Signaling, 9452). Western blot results shown are representative blots from three experiments. Band intensities were quantified using the Biorad Image Lab software Version 6.1.0 build 7.

### 2.6. Cell Cycle Assays

MB468, R8, HEK293T, Res1 (R1) cells were all transduced with Incucyte Cell Cycle Green/Red Lentivirus Reagent (Sartorius, Göttingen, Germany, 4779) according to the manufacturer’s instructions. Cells were synchronized using 2.5 mM thymidine (Sigma Aldrich, T1895) for 24 h and were then treated with either methionine or −Met+Hcy media. The Sartorius IncuCyte S3 Live Cell Analysis System was used to capture images and analyze data.

### 2.7. Cell Death Assay

Cells were grown overnight in complete DMEM media and then placed in a 96-well plate at a density of 4000 cells/well with their respective media. Incucyte^®^ Cytotox Red Dye (Sartorius, 4632) was added according to the manufacturer’s instructions to measure cell death. Incucyte^®^ Annexin V Green Dye (Sartorius, 4642) was added according to the manufacturer’s instructions to measure apoptosis. The Sartorius IncuCyte S3 Live Cell Analysis System was used to capture images and analyze data.

### 2.8. RNA Sequencing

MB468 and R8 samples were collected as reported in Borrego et al. [49]. HEK293T and R1 cells were cultured in either +Met or −Met+Hcy media for 12 h, and total RNA was extracted using a kit (Qiagen, Hilden, Germany, 74104). Samples were then sent to Novogene Corporation Inc. for library preparation and sequencing (Sacramento, CA, USA).

Once obtained, sequencing reads were quantified and aligned to the human genome GRCh38 using kallisto v0.48.0 [50]. Estimated counts of transcripts from the same Ensembl gene ID were added together, and the sums were converted to integers. Genes with a sum of counts less than 10 were excluded from the analysis. Differential expression analysis was performed using DESeq2 v1.38.3 [51].

Volcano plots were generated using ggplot2 v3.4.4 [52]. Genes with log fold changes greater than 1 and adjusted *p*-values less than 0.05 were considered significant.

Venn diagrams depicting overlapping genes with adjusted *p*-values less than 0.05 were plotted using eulerr v7.0.1 [53].

Enrichment analysis using GO Biological Process 2021 was performed using clusterProfiler v4.6.2 [54,55].

UpSet plots were generated using UpSetR v1.4 [56].

### 2.9. Metabolite Measurements Using LC-MS

To measure SAM and SAH concentrations, cultured cells were switched to −Met+Hcy media. After incubation in a CO_2_ incubator, metabolites were extracted from 5 × 10^6^ cells per plate using 1 mL of ice-cold 80% MeOH for 15 min at −80 °C. Cells were collected by rubber policemen and transferred to 1.5 mL Eppendorf tubes on dry ice. The samples were homogenized by vortexing for 10 s. The samples were then centrifuged at 16,000× *g* for 10 min at 4 °C, and 40 μL of supernatant was loaded into individual LC-MS vials. Metabolites were analyzed by a quadrupole-orbitrap mass spectrometer (Q-Exactive Plus Hybrid Quadrupole-Orbitrap, Thermo Fisher, Waltham, MA, USA) coupled to Vanquish UHPLC Systems (Thermo Fisher) via electrospray ionization. LC separation was on an Xbridge BEH amide column (2.1 mm × 150 mm, 2.5 µm particle size, 130 Å pore size; Waters, Milford, MA, USA) at 25 °C using a gradient of solvent A (5% acetonitrile in water with 20 mM ammonium acetate and 20 mM ammonium hydroxide) and solvent B (100% acetonitrile). The flow rate was 150 µL/min. The LC gradient was: 0 min, 90% B; 2 min, 90% B; 3 min, 75% B; 7 min, 75% B; 8 min, 70% B; 9 min, 70% B; 10 min, 50% B; 12 min, 50% B; 13 min, 25% B; 14 min, 20% B; 15 min, 20% B; 16 min, 0% B; 20.5 min, 0% B; 21 min, 90% B; 25 min, 90% B. Autosampler temperature was set at 4 °C and the injection volume of the sample was 15 μL. MS analysis was acquired in positive ion modes with full MS scan mode (SIM) from *m*/*z* 70 to 830 and 140,000 resolution. The concentrations of SAM and SAH were measured with authentic chemical standards. Peaks that matched SAM and SAH standards were identified with El-Maven v0.12.0.

### 2.10. Autophagy

Cells were plated in DMEM media and switched to either +Met, −Met+Hcy (3 h or 24 h), +Met+100 nM Rapamycin for 24 h (Medchem Express, Monmouth Junction, NJ, USA, HY-10219). 10 nM of Bafilomycin 1 A (Sigma Aldrich, SML 1661) was added for the last hour of treatment. Cells were harvested and lysed using RIPA Buffer (150 mM sodium chloride, 50 mM Tris pH8, 1% NP40, 0.05% sodium deoxycholate, 0.1% SDS). LC3 was measured using western blot.

### 2.11. Activity Assay

HEK293T cells were transiently transfected with plasmid encoding HA-tagged PP2A catalytic subunit (HA-PP2Ac) using Lipofectamine 2000. Twenty-four hours post-transfection, cells were rinsed twice in ice-cold PBS and lysed in 50 mM Tris·HCl pH 7.5, 150 mM NaCl, 1% NP-40, 1 mM EDTA plus protease inhibitors. Lysates were cleared at 14,000× *g* for 10 min at 4 °C and quantified by BCA assay. 500 µg lysate was incubated with 30 µL anti-HA magnetic beads (Pierce, Appleton, WI, USA) for 2 h at 4 °C. Beads were washed 4 times in lysis buffer and twice in reaction buffer (50 mM Tris·HCl pH 7.5, 1 mM DTT, 0.1 mM EDTA). A subset of bead-bound IPs was pre-incubated with 1 µM okadaic acid (OA) for 15 min at 4 °C as a negative control. Beads were resuspended in 50 µL reaction buffer containing 5 mM p-nitrophenyl phosphate (pNPP; Sigma N7653) and incubated at 30 °C for 30 min. Reactions were stopped by adding 50 µL 1 M NaOH. Absorbance at 405 nm was measured on a plate reader. A 10 mM pNP stock was serially diluted twofold to give nine standards (0–320 µM). 50 µL of each was mixed with 50 µL 1 M NaOH, incubated for 5 min at 25 °C, and OD_405_ recorded. The calibration curve (OD_405_ vs. [pNP]) was linear over the full range (R^2^ = 0.998). Protocol was adapted from Fellner et al. 2003 [57].

### 2.12. Methyl-Proteome Profiling

#### Lysate Preparation

Cell lysates were resuspended in 50 mM Tris pH 7.5, 8 M urea, 1 mM activated sodium vanadate, 2.5 mM sodium pyrophosphate, 1 mM β-glycerophosphate, and 100 mM sodium phosphate. One milligram of protein for SCX samples and 100 µg for whole cell lysates were measured by BCA assay. The lysates went through sonication and high-speed centrifugation, followed by filtration through a 0.22 μm filter. 5 mM DTT, 25 mM iodoacetamide, and 10 mM DTT were used to reduce, alkylate, and quench protein, respectively. Lysates were diluted 4-fold in 100 mM Tris pH 8 and 1:100 dilution of trypsin was used to digest proteins. 5% TFA was used to quench samples and reduced pH 2. For 1 mg SCX samples, peptides were purified by using reverse-phase Sep-pak C18 Cartridges (Waters) and eluted with 50% acetonitrile and 0.1% TFA. For WCL samples, peptides were purified by a C18 stage tip created in-house. Purified samples were dried by speed vac. WCL samples were ready to inject into LC-MS and 1 mg samples went for SCX enrichment.

Dried 1 mg samples were resuspended in 60% acetonitrile and 40% BRUB (5 mM phosphoric acid, 5 mM boric acid, 5 mM acetic acid, pH 2.5) and incubated with high pH SCX beads (Sepax, Newark, DE, USA) for 30 min. After washing with washing buffer (80% acetonitrile, 20% BRUB, pH 9), four fractions were eluted with the following buffer compositions: elution buffer 1 (60% acetonitrile, 40% BRUB, pH 9), elution buffer 2 (60% acetonitrile, 40% BRUB, pH 10), elution buffer 3 (60% acetonitrile, 40% BRUB, pH 11), elution buffer 4 (30% acetonitrile, 70% BRUB, pH 12). Eluted fractions were dried in a speedvac and resuspended in 1% TFA and an in-house stage tip consisting of 2 mg HLB beads (Waters) and 1 layer of C8 was used to desalt the samples [58].

### 2.13. Mass Spectrometric Analysis for 3 Hr Met vs. Hcy Treatment

All liquid chromatography (LC)–MS experiments were performed on a nanoscale UHPLC system (EASY-nLC1200, Thermo Scientific) connected to a Q Exactive Plus hybrid quadrupole-Orbitrap mass spectrometer equipped with a nanoelectrospray source (Thermo Scientific). Peptides were separated by a reversed-phase analytical column (PepMap RSLC C18, 2 μm, 100 Å, 75 μm × 25 cm) (Thermo Scientific). For high pH SCX fractions, the flow rate was set to 300 nL/min at a gradient starting with 0% buffer B (0.1% FA, 80% acetonitrile) to 29% B in 142 min, then washed by 90% B in 10 min, and held at 90% B for 3 min. The maximum pressure was set to 1180 bar, and the column temperature was constant at 50 °C. Dried SCX fractions were resuspended in buffer A and injected as follows: E1: 3 μL of 60 μL; E2-4: 5 μL of 6 μL. Peptides separated by the column were ionized at 2.0 kV in the positive ion mode. MS1 survey scans for data-dependent acquisition were acquired at a resolution of 70 k from 350 to 1800 *m*/*z*, with a maximum injection time of 100 ms and an automatic gain control (AGC) target of 1 × 10^6^. MS/MS fragmentation of the 10 most abundant ions was analyzed at a resolution of 17.5 k, AGC target of 5 × 10^4^, and a maximum injection time of 240 ms, and normalized collision energies of 32 were used. Dynamic exclusion was set to 30 s and ions with charge 1 and >6 were excluded. For WCL samples, the Q Exactive Plus was operated in positive-ion nano-electrospray mode (2.0 kV) and acquired one full MS scan (*m*/*z* 350–1800; 70,000 resolution at *m*/*z* 200; AGC target 1 × 10^6^; max IT 100 ms), followed by 32 data-independent MS/MS windows spanning *m*/*z* 350–1800 (variable widths; 35,000 resolution; AGC target 3 × 10^6^; max IT “Auto”; HCD NCE 26; MSX isochronous injection on). All scans were collected in profile mode without dynamic exclusion. The DIA window scheme was adapted from Bruderer et al. [59] to yield approximately eight data points per chromatographic peak over a two-hour gradient.

### 2.14. Identification of Peptides

Proteome Discoverer SEQUEST (version 2.2, Thermo Scientific) was used to search MS/MS fragmentation spectra against the Uniprot Homo sapiens database with all reviewed isoforms (Downloaded August 2022). For SCX samples, the maximum missed cleavage rate was set to 4, and for WCL samples, it was set to 2. Trypsin was set to cleave at R and K. Dynamic modifications were set to include monomethylation of arginine or lysine (R/K, +14.01565), dimethylation of arginine or lysine (R/K, +28.0313), trimethylation of lysine (K, +42.04695), oxidation on methionine (M, +15.995 Da), and acetylation on protein N-terminus (+42.011 Da). Fixed modification was set to carbamidomethylation on cysteine residues (C, +57.021 Da). The maximum parental mass error was set to 10 ppm, and the MS/MS mass tolerance was set to 0.02 Da. Peptides with a sequence of 7–50 amino acids were considered. For SCX samples, methylation site localization was determined by the ptm-RS node in Proteome Discoverer, and only sites with localization probability greater than or equal to 75% were considered. DIA data were processed in DIA-NN v1.8.1 [60] using an in silico spectral library, considering peptides of 7–50 amino acids and allowing up to one missed cleavage.

### 2.15. Methyl False Discovery Estimation

PSM and decoy PSM files were exported from Proteome Discoverer 2.2 and filtered to include only methylated and decoy PSMs. As previously described [61], the false discovery rate (FDR) was calculated, and target methyl PSMs were removed until a 1% FDR threshold was achieved.

### 2.16. Bioinformatic Analysis

Peptide group intensities exported from Proteome Discoverer 2.2 were first filtered to retain only features with at least one non-zero value per condition. Missing values were then classified as “missing at random” (MAR), peptides with a maximum of one missing value in each group, or “missing not at random” (MNAR), peptides detected consistently in one condition but absent in the other. MAR values were imputed using a truncated-normal nearest-neighbor algorithm (KNN-TN) implemented in R [62] and then combined with peptides with nonrandom missing values. MNAR values were imputed in Perseus v2.0.6.0 by drawing from a left-shifted Gaussian distribution as described by [63]. After imputation, peptide intensities were normalized by VSN normalization. For protein-level summarization, we applied the Deffacto algorithm with a minor library amendment to extract sample-level abundances and incorporated the PECA framework for peptide-to-protein roll-up [64], yielding *p*-values that were adjusted to q-values via the Benjamini–Hochberg procedure. Methyl-proteomics data underwent the same MAR/MNAR imputation and isoform curation; only peptide groups with at least one PSM passing target–decoy FDR (<1%) were retained. For peptides mapping to proteins quantified in the WCL dataset, PTM-specific abundance changes were modeled in MSstatsPTM [65], which integrates modified and unmodified signals to compute PTM *p*-values. Peptides without WCL protein quantification were analyzed by Welch’s *t*-test. For peptides detected in more than one fraction, the peptide with the highest abundance was selected. All resulting p-values were corrected for multiple hypothesis testing using the Benjamini–Hochberg method.

## 3. Results

### 3.1. Characterization of the Cell Cycle and Survival Response of Methionine-Dependent and Independent Cells

Most cancer cells exhibit a strict dependence on exogenous methionine and are unable to proliferate when methionine is replaced with its precursor homocysteine. In contrast, non-tumorigenic cells are generally able to proliferate under these nutrient conditions [12,13,14]. Consistent with this, neither the triple-negative breast cancer cell line MDA-MB-468 (Figure 1B) nor the HEK293T cell line Appendix A proliferates in methionine-free, homocysteine-supplemented (−Met+Hcy) media. Previous studies have described selection schemes to isolate clones from methionine-dependent cancer cell lines that acquire the ability to grow in homocysteine-containing media [47,48]. This phenotypic switch to methionine independence is often associated with a loss of tumorigenic potential [47,48,66].

We previously reported the generation of methionine-independent clones from MDA-MB-468 (MB468-R8) [18] and here we developed methionine-independent clones from HEK293T cells, we refer to these cells as 293Tres-R1. Both clones exhibit robust proliferation in −Met+Hcy media at rates comparable to those observed in complete medium (Figure 1B and Appendix A). To characterize their cell cycle behavior under these conditions, we performed live-cell imaging using TagGFP2- and mKate2-tagged Geminin and Cdt1, respectively [67]. Cells were synchronized in early S-phase via thymidine block and subsequently released into either +Met-Hcy (complete) or −Met+Hcy media. As expected, parental MB468 cells progressed through the cell cycle in complete medium (Figure 1C). However, upon release into −Met+Hcy medium, most MB468 cells completed S-phase and mitosis but exhibited a strong delay or arrest at the G1-to-S transition (Figure 1C). In contrast, the R8 clone continued to cycle in both media conditions (Figure 1D). A similar pattern was observed for the HEK293T and 293Tres-R1 pair. Parental HEK293T cells rapidly arrested after release into −Met+Hcy medium, while 293Tres-R1 continued normal proliferation Appendix A. Interestingly, the nature of the cell cycle arrest differed between the two parental lines. MDA-MB-468 cells completed S-phase and mitosis and arrested at the subsequent G1/S transition, while HEK293T cells failed to complete the cell cycle and arrested in S/G2 phase. These differences may reflect the distinct origins and genetic backgrounds of the cell lines. MDA-MB-468 is derived from a triple-negative breast carcinoma, whereas HEK293T cells are adenovirus 5–transformed human embryonic kidney cells that express SV40 large T antigen. The latter inactivates the tumor suppressors RB and p53 [68,69], and has been shown to cause an arrest in late S/G2 in response to methionine restriction [70]. This likely contributes to the observed differences in the cell cycle response to methionine restriction between HEK293T and MDA-MB-468 cells. An S/G2 arrest has previously also been observed in response to methioninase treatment, which depletes methionine in growth media [71]. Of note, methionine-independent cells are more sensitive to methioninase treatment compared to normal cells, and are thus still considered methionine addicted [72].

Visual inspection of live-cell imaging data suggested progressive cell death in methionine-dependent cells cultured in −Met+Hcy medium. To investigate whether this was due to apoptosis, we cultured both MDA-MB-468 and R8 cells in −Met+Hcy medium and monitored cleavage of the apoptotic markers PARP and Caspase-3 by immunoblotting. As early as two days after the switch to −Met+Hcy conditions, we observed increased PARP cleavage in MB468 cells, indicating the onset of apoptosis (Figure 1E). By day four, both cleaved PARP and cleaved Caspase-3 were markedly elevated. In contrast, R8 cells maintained stable PARP levels throughout the experiment, suggesting that methionine-independent cells do not activate apoptosis under these conditions (Figure 1E).

As a positive control, treatment with 1 mM doxorubicin (DOX) robustly induced apoptotic marker cleavage in MB468 cells Appendix A. To rule out the possibility that R8 cells harbor mutations or epigenetic changes impairing apoptotic signaling, we determined the IC_50_ values for DOX in both cell lines Appendix A. The IC_50_ values were comparable, confirming that R8 cells retain intact apoptotic machinery but, unlike the parental line, do not undergo apoptosis in response to methionine restriction.

To monitor apoptosis in real time, we used Annexin V staining and live-cell imaging. Consistent with the immunoblotting results, MB468 cells exposed to −Met+Hcy medium exhibited a marked increase in Annexin V signal, starting around day two of treatment and continuing thereafter (Figure 1F). In contrast, HEK293T and R1 cells displayed only modest accumulation of cleaved PARP and cleaved Caspase-3 after five days in −Met+Hcy, suggesting limited activation of apoptotic pathways Appendix A. Notably, PARP cleavage was also minimal in the DOX-treated HEK293T cells, indicating an intrinsic resistance to apoptosis in this cell line Appendix A. Consistent with this, the DOX IC_50_ was substantially higher in HEK293T than in MB468 cells Appendix A. Interestingly, R1 cells, despite being more sensitive to DOX-induced apoptosis than parental HEK293T cells, did not show increased PARP or Caspase-3 cleavage when grown in homocysteine medium Appendix A. Live-cell Annexin V imaging further confirmed that HEK293T cells undergo limited apoptosis in −Met+Hcy medium compared to MB468 cells Appendix A, although significantly more than R1 cells Appendix A. However, when overall cell death was assessed using a Cytotox dye, HEK293T and MB468 cells showed comparable levels of cell death in −Met+Hcy Appendix A, suggesting that HEK293T cells may engage a non-apoptotic form of cell death in response to methionine depletion.

### 3.2. Methionine-Independent Cells Likely Employ Distinct Adaptive Mechanisms

To investigate the transcriptional response to methionine depletion, we performed RNA sequencing on methionine-dependent and -independent cell lines after 12 h of growth in −Met+Hcy medium. Both methionine-sensitive lines, MB468 and HEK293T, exhibited robust transcriptional changes: 620 genes were upregulated and 620 downregulated in MB468 cells, while 881 genes were upregulated and 497 downregulated in HEK293Ts (Figure 2A and Appendix A). A total of 189 upregulated and 102 downregulated genes were commonly regulated in both MB468 and HEK293T cells. While there was substantial overlap, many differentially expressed genes were unique to each cell line, indicating both shared and cell line-specific responses (Figure 2A and Appendix A). Gene ontology analysis highlighted pathways related to transcription and lipid metabolism as significantly enriched, consistent with the physiological phenotypes observed (Figure 2B).

Methionine-independent cell lines also mounted transcriptional responses when grown in −Met+Hcy medium. R8 cells displayed 81 upregulated and 397 downregulated genes, whereas R1 cells showed more upregulation, with 113 upregulated and only 25 downregulated genes (Figure 2C and Appendix A). Strikingly, the two methionine-resistant lines shared only three overlapping differentially expressed genes: OSMR, EIF3CL, and IGFBP3 (Figure 2C). These genes do not share clear functional roles in methionine sensing or metabolism and are therefore unlikely to represent a shared resistance program. Instead, the data suggest that methionine independence arises through distinct, cell line-specific adaptive mechanisms. The UpSet plot analysis of up- and down-regulated genes (Figure 2D,E) offers a more detailed view of our transcriptional changes. It highlights the large set of genes commonly altered in the two methionine-dependent parental lines, while clearly illustrating the minimal intersection between the resistant clones, underscoring that each likely acquired methionine independence via distinct mechanisms. We also compared the transcriptional responses of methionine-dependent and -independent lines within the isogenic MB468/R8 and HEK293T/R1 cell pairs Appendix A. In the MB468/R8 pair, around 40% of the upregulated genes in R8 overlapped with those upregulated in MB468, while almost 60% of the downregulated genes were shared Appendix A. In contrast, the HEK293T/R1 comparison revealed a higher degree of similarity, with roughly two-thirds of upregulated and half of downregulated genes overlapping Appendix A. Gene ontology analysis identified lipid metabolism as the most significantly enriched shared pathway in the MB468/R8 comparison Appendix A, while transcriptional regulation and necrotic cell death pathways were enriched among the overlapping genes in the HEK293T/R1 pair Appendix A. The response in lipid metabolism found in the MB468 background is consistent with metabolomics data, which demonstrated significant remodeling of lipids when cells were shifted to −Met/+Hcy medium [49]. These results underscore that these isogenic systems exhibit distinct adaptive transcriptional programs in response to homocysteine growth conditions, which is further supported by the distinct scattering seen in the PCA plot (Figure 2F).

### 3.3. Carboxy-Methylation of the Catalytic PP2A Subunit Is Highly Responsive to Changes in Exogenous Methionine

As previously reported, S-adenosylmethionine (SAM) synthesis declines under methionine-limiting conditions, and supplementation with SAM can rescue the growth of MDA-MB468 cells cultured in −Met+Hcy medium [23]. Since R8 cells continue to proliferate under these same conditions, we hypothesized that they might maintain higher SAM levels compared to the methionine-dependent MB468 cells. To test this, we measured intracellular SAM and S-adenosylhomocysteine (SAH) levels over time in MB468 and R8 cells cultured in either a complete or −Met+Hcy medium (Figure 3A and Appendix A). Surprisingly, both cell lines showed a rapid and substantial decrease in SAM levels, dropping below 5 µM, accompanied by a sharp reduction in methylation potential, as measured by the SAM/SAH ratio. Interestingly, baseline SAM levels were significantly lower in R8 cells (~40 µM) than in MB468 cells (~100 µM) (Figure 3A vs. Appendix A).

We observed a similar pattern in the HEK293T and methionine-independent R1 cells Appendix A: both showed a marked drop in SAM levels. However, in contrast to the MB468/R8 comparison, R1 cells did not have significantly different baseline levels of SAM, although both were substantially lower than the SAM levels in MB468 and R8 cells.

Given the substantial drop in SAM levels, we expected widespread changes in the protein methylome, as SAM is the primary methyl donor in cells. To assess this, we performed global protein methylation profiling in MB468 cells using liquid chromatography-mass spectrometry-based (LC-MS) proteomics [73]. Despite the significant reduction in methylation potential after just 3 *h* in −Met+Hcy medium, none of the 528 lysine or arginine methyl-peptides quantified by LC-MS were significantly changed in abundance (Figure 3B, Appendix A).

Given the unexpected result that global protein methylation remained unchanged after 3 h of methionine deprivation, we hypothesized that methionine-responsive methylation might occur on specific, non-canonical protein targets. In the budding yeast model, methionine starvation is well characterized, with cell cycle progression tightly coordinated with methionine availability through the SCF-Met30 ubiquitin ligase complex [32,33,34,35,74]. Notably, in yeast, the catalytic subunit of protein phosphatase 2A (PP2A) undergoes carboxy-terminal methylation, which is highly sensitive to methionine levels. Normally, this methylation suppresses autophagy, and when it is lost during methionine starvation, autophagy is induced to recycle intracellular proteins for methionine generation via lysosomal degradation [36].

Given the high degree of conservation of the PP2A carboxy-terminal methylation machinery between yeast and humans, we investigated whether PP2A methylation changes occur in response to a shift of cells to −Met+Hcy medium in human cells. Remarkably, within just 2 h of shifting methionine-dependent MB468 cells, we observed a substantial reduction in methylated PP2Ac and a corresponding accumulation of the demethylated form (Figure 3C–E). In contrast, the methionine-independent R8 cells maintained stable PP2Ac methylation throughout the time course (Figure 3E), highlighting a strong association between PP2Ac methylation status, methionine metabolism, and proliferative capacity.

A similar pattern was observed in the HEK293T/Res1 (R1) cell pair: a shift to −Met+Hcy medium led to rapid demethylation of PP2Ac in the methionine-sensitive HEK293T cells, whereas the methionine-independent R1 cells retained methylation Appendix A. Interestingly, despite exhibiting distinct transcriptional responses when cultured in −Met+Hcy medium, both resistant cell lines preserved PP2Ac methylation.

Cancer cells, which are naturally methionine-independent (pancreatic cancer line PANC1 and breast cancer line MDA-MB231), only showed a transient and modest increase in demethylated PP2A after 3 h of growth in −Met+Hcy, with levels returning to baseline by 24 h (Figure 4A,B). In contrast, the methionine-dependent pancreatic cancer line BxPC3 maintained elevated levels of demethylated PP2A (Figure 4C).

These results suggest that carboxy-methylation of PP2Ac may serve as a conserved, early sensor of methionine availability and cellular methylation potential, acting upstream of downstream responses that define methionine dependence of cancer.

### 3.4. Methionine-Dependence Is Not Mediated Through mTORC1 Signaling

Cells rely on nutrient-sensing pathways to adapt to environmental changes, and the mTORC1 pathway is a central regulator linking nutrient availability to cell growth and proliferation [75]. Previous studies identified SAMTOR as a sensor of S-adenosylmethionine (SAM) that connects SAM levels to mTORC1 activity [37,38]. To investigate whether the proliferation defect in methionine-dependent MB468 cells involves mTORC1 signaling, we assessed S6 phosphorylation, a downstream target of mTORC1, as well as PP2A methylation under conditions of complete methionine starvation (without homocysteine supplementation).

As expected and consistent with prior reports [37,38], S6 phosphorylation was significantly reduced under methionine starvation, indicating mTORC1 inhibition (Figure 5A). Leucine depletion and rapamycin treatment served as positive controls. Notably, leucine depletion failed to inhibit mTORC1 signaling in MB468 cells, consistent with previous findings [76].

Surprisingly, when methionine was replaced with homocysteine (−Met+Hcy), mTORC1 activity remained sustained over a 3-day period, as evidenced by stable phosphorylation levels of both S6 and 4EBP1 (Figure 5B). This persistent activity was unexpected, as intracellular SAM levels in MB468 cells dropped below 5 µM within just 3 h under −Met+Hcy conditions (Figure 3A and Appendix A). The dissociation constants (K_d_) of SAMTOR for SAM and SAH have been reported as 7 µM and 4 µM, respectively [38,77]. SAM binding to SAMTOR triggers an allosteric conformational change that disrupts its interaction with GATOR1, thereby allowing mTORC1 activation. Notably, this mechanism is not SAM-specific. SAH can also induce the same allosteric switch, enabling SAMTOR to maintain mTORC1 activity in low-SAM, high-SAH environments [38,77]. Given that SAH levels in −Met+Hcy-treated cells exceed 10 µM, well above the K_d_ for SAMTOR, the sensor is likely SAH-bound under these conditions, which could account for the maintained mTORC1 signaling. In support of this possibility, homocysteine has been reported to activate mTORC1 in neuronal cells [77], potentially through elevated SAH levels and enhanced SAMTOR dissociation from GATOR1. Additionally, we found no significant autophagy induction in the MB468 cells with either −Met+Hcy treatment or PP2A demethylation through overexpression of the Protein Phosphatase Methylesterase-(PME-1) (Figure 5C).

### 3.5. PP2Ac Methylation State Controls Methionine Dependence

Under normal growth conditions, approximately 90% of PP2A exists in its methylated form [44]. Given our earlier findings that loss of PP2Ac methylation correlates with proliferation defects under methionine-depleted (+homocysteine) conditions, we hypothesized that forced demethylation of PP2Ac in methionine-independent cells might render them methionine-dependent.

To test this, we generated stable cell lines overexpressing Protein Phosphatase Methylesterase-1 (PME-1), a known demethylase of PP2Ac [78,79,80]. PME-1 overexpression in MB468 cells produced the expected increase in demethylated PP2Ac (Figure 6A), with no further effect on proliferation in the already growth-inhibited state under −Met+Hcy conditions. Unexpectedly, PME-1 expression slightly enhanced MB468 proliferation in standard medium. Similarly, R8 cells showed modest growth enhancement in complete medium. However, critically, PME-1 overexpression induced methionine dependence in the otherwise methionine-independent R8 cells (Figure 6A).

Comparable effects were observed in the HEK293T/Res1 model (Figure 6B). PME-1 overexpression led to complete methionine dependence in R1 cells, without significantly affecting their growth in methionine-replete conditions. While PP2A is a major substrate of PME-1, this enzyme can also demethylate other members of the PPP family, such as PP4c [78,79,80,81], which could contribute to the observed phenotypes.

To directly test whether PP2Ac demethylation alone is sufficient to alter methionine sensitivity, we expressed a Leucine309 deletion mutant (∆Leu309) of PP2Ac in R8 and R1 cells in addition to the endogenous PP2Ac. Leu309 is the C-terminal residue required for carboxy-methylation, and its deletion mimics a demethylated state [81]. In R8 cells, the expression of PP2Ac-∆Leu309 resulted in significantly impaired proliferation compared to parental controls, despite intact methylation of endogenous PP2Ac. This growth defect was only observed in −Met+Hcy growth conditions, indicating that ∆Leu309 expression sensitizes cells to reduced methionine availability rather than causing toxicity on its own Appendix A. Notably, individual ∆Leu309 clones with higher transgene expression showed a more severe growth defect, indicating a dose-dependent dominant effect (Figure 6C).

In R1 cells, expression of PP2Ac-∆Leu309 also induced methionine dependence in a dominant manner (Figure 6D). Unlike the dose-dependent effects observed in R8 cells, we did not detect a graded response in R1 cells. This may be due to the heightened sensitivity of R1 cells to PP2Ac methylation status, as evidenced by the complete proliferation block following PME-1 overexpression (Figure 6B). Even low levels of PP2Ac-∆Leu309 expression were sufficient to significantly impair proliferation in −Met+Hcy medium, underscoring the critical role of PP2Ac methylation in maintaining methionine independence in this cell type.

Collectively, these findings provide strong evidence that PP2Ac methylation directly regulates methionine dependence in cancer cells. Demethylation of PP2Ac is sufficient to convert methionine-independent cells into methionine-dependent ones, highlighting the central role of this post-translational modification in linking nutrient sensing to proliferative capacity. The growth defect observed in PME-1 overexpression and ∆Leu methionine-independent cells following −Met/+Hcy treatment is likely due to a cell cycle arrest, similar to what is seen in their methionine-dependent counterparts, but this remains to be elucidated.

## 4. Discussion

The dependence of cancer cells on exogenous methionine is a long-recognized but mechanistically unresolved metabolic vulnerability. Our study aimed to uncover a molecular link between methionine availability and cell proliferation. We identify the catalytic subunit of Protein Phosphatase 2A (PP2Ac), which undergoes carboxy-methylation on its terminal residue, leucine 309, as a key effector in this process. We show that in methionine-dependent cells, PP2Ac rapidly loses its methylation upon methionine depletion and homocysteine supplementation. In contrast, methionine-independent cells maintain PP2Ac methylation under identical conditions for several days (Figure 3). Strikingly, this differential PP2Ac methylation response occurs despite a comparable and rapid reduction in S-adenosylmethionine (SAM) levels and SAM/SAH ratios across all cell lines when shifted to –Met+Hcy medium. This raises the intriguing question of how methionine-independent cells preserve PP2Ac methylation despite a compromised cellular methylation potential. One possibility is that local pools of SAM, rather than global methylation status, sustain PP2Ac methylation in these cells.

Under basal conditions, methionine-independent R8 cells exhibit significantly lower SAM levels, less than 50% of those in their methionine-dependent parental MB468 cells (Figure 3A), suggesting they may be pre-adapted to tolerate methionine depletion. In contrast, methionine-independent R1 cells maintain basal SAM levels comparable to their HEK293T parental line Appendix A, though these levels are approximately fivefold lower than those observed in the MB468 background. Despite these distinct metabolic profiles and divergent transcriptional responses (Figure 2), both R8 and R1 cells are able to proliferate in the absence of exogenous methionine. These observations suggest that methionine independence arises through different strategies of SAM homeostasis, ultimately converging on the shared ability to sustain PP2Ac methylation under methionine-restricted conditions.

Our methyl-proteome profiling, based on strong cation exchange (SCX) chromatography and trypsin digestion, selectively detects canonical methylation on lysine and arginine residues. Trypsin cleaves the C-terminal side of either lysine or arginine, however, methylation of these residues inhibits cleavage, resulting in a higher charge that can be selected with this strategy [73]. Non-canonical modifications like the methylation of PP2Ac are not captured by this approach. While other methylation targets likely contribute to methionine dependence, our functional studies strongly implicate PP2Ac methylation as a central node. Forced demethylation either via overexpression of PME-1 or genetic deletion of Leu309, was sufficient to convert methionine-independent cells into methionine-dependent ones (Figure 6), underscoring the causal and predictive role of PP2Ac methylation in linking methionine metabolism to proliferation capacity. Additional support for this role of PP2A comes from budding yeast, where Sutter et al. (2013) showed that PP2Ac methylation integrates methionine metabolism with autophagy regulation [36]. In contrast, our data in the mammalian systems does not reveal a strong connection between PP2Ac methylation and autophagy (Figure 5C). Instead, PP2Ac methylation appears to directly couple methionine availability with cell proliferation and survival. This function resembles the role of the SCF-Met30 ubiquitin ligase in yeast, where methionine metabolism regulates cell cycle inhibitors and transcriptional regulators to coordinate cell proliferation response with methionine metabolism [30,32,33,82].

The mTORC1 complex is a central regulator that integrates metabolic inputs to control cell growth, proliferation, and homeostasis [83,84]. A sophisticated mechanism has been described that links methionine metabolism and intracellular SAM levels to mTORC1 activation via SAMTOR [37,38]. Under conditions of methionine sufficiency, SAM binds to SAMTOR, preventing its interaction with GATOR1. This opens a binding site on GATOR1 for PRMT1, allowing PRMT1 to methylate the GATOR1 subunit NPRL2, thereby inhibiting GATOR1’s GAP activity and promoting mTORC1 signaling. When SAM levels drop, SAM dissociates from SAMTOR and binds GATOR1, displacing PRMT1, preventing NPRL2 methylation, and leading to inhibition of mTORC1 [37,38,85]. Consistent with this model, we observed that complete methionine depletion in MDA-MB468 cells leads to robust mTORC1 inhibition, as indicated by reduced phosphorylation of downstream targets (Figure 5A). However, in striking contrast, supplementation of methionine-depleted media with homocysteine sustains full mTORC1 signaling over several days (Figure 5B), despite SAM levels falling below the SAMTOR dissociation constant of ~7 µM within 3 hours in −Met+Hcy medium (Figure 3A). This can be reconciled by the observation that SAH, present at high concentrations under these conditions, also binds SAMTOR (Kd ~4 µM) and promotes its dissociation from GATOR1, thereby maintaining mTORC1 activation even under −Met+Hcy conditions [37,38,85].

While the loss of SAMTOR-mediated inhibition due to the presence of SAH can partially explain sustained mTORC1 signaling, it remains unclear how the pathway bypasses the requirement for PRMT1 activity. Under −Met+Hcy conditions, PRMT1 would be expected to lose activity due to insufficient SAM (PRMT1 Kd ~26 µM [37,86]) and the accumulation of SAH, a known inhibitor of PRMT1 [87]. One possible explanation is that localized SAM concentrations at the lysosomal surface may remain sufficient to enable NPRL2 methylation and thus PRMT1 function. Alternatively, PRMT1-dependent methylation of NPRL2 may not be required under these conditions, and an as-yet-unidentified compensatory mechanism may maintain mTORC1 activity. These results are consistent with a previous report in mouse cells that showed that methionine dependence is independent of mTORC1 signaling [29]. Regardless of the precise molecular details, our data clearly show that mTORC1 activity is not compromised in MB468 cells grown in −Met+Hcy medium. This observation, combined with our functional experiments, strongly argues that mTORC1 signaling is not the primary driver of methionine dependence in cancer cells. Instead, our findings highlight the importance of alternative signaling pathways, such as PP2A methylation, in mediating this metabolic vulnerability.

The methylation of PP2Ac occurs within the highly conserved TPDYFL motif at its C-terminus. This region is subject to multiple posttranslational modifications, including carboxyl-methylation of Leucine 309 [88]. Notably, this methylation event has been linked to changes in both PP2A phosphatase activity and its association with regulatory B subunits or other interaction partners [44,57,81,88]. While our data show that demethylation of PP2Ac in response to methionine depletion does not alter intrinsic phosphatase activity Appendix A, it is likely that demethylation impacts PP2A holoenzyme composition, subcellular localization, or substrate specificity, leading to downstream effects on cell proliferation and survival. Importantly, our activity assay measures only general phosphomonoester hydrolysis on a non-proteinaceous substrate and does not capture the dynamic and substrate-specific changes in phosphatase activity that may occur in vivo. It is therefore plausible, and even likely, that PP2Ac methylation modulates phosphatase activity toward specific substrates through mechanisms involving altered B subunit assembly, spatial localization, or sequence-context-dependent interactions.

Importantly, our results demonstrate that loss of PP2Ac methylation is necessary for the observed proliferation defect in methionine-dependent cancer cells, specifically when grown in −Met+Hcy medium. However, this effect is context dependent. For example, demethylation induced by PME-1 overexpression does not impair proliferation in the standard medium and may even promote growth (Figure 6). These findings suggest that metabolic context, specifically methionine depletion and altered methylation potential, cooperates with PP2Ac demethylation to regulate cell fate decisions.

Our experiments were designed to capture early cellular responses to methionine depletion, with a particular focus on identifying sensing mechanisms. Among these, PP2Ac carboxy-methylation emerged as a rapid and sensitive responder to changes in methylation potential, positioning it as a likely early sensor of methionine/SAM availability. In contrast, most other methylation events remained unchanged within the initial 3-h window, indicating that alterations to global methylation patterns may occur later in response to prolonged methionine restriction. Nonetheless, several methylation-dependent processes are ultimately impacted under methionine-limited conditions. For instance, histone methylation marks are known to change over time [89], and PRMT5-mediated methylation of Sm proteins is reduced, resulting in widespread splicing defects in cells cultured in −Met+Hcy medium [39]. Similarly, reduced SAM levels have been shown to impair RIPK1 methylation, which has been linked to inflammatory signaling and apoptosis [90]. While these later effects likely contribute to the broader physiological adaptation to methionine restriction, the rapid demethylation of PP2Ac stands out as a candidate early event that may transduce metabolic status into signaling pathways governing cell proliferation and survival.

The precise mechanisms by which PP2Ac methylation controls proliferation in cancer cells, and its effects on cell cycle progression and viability, remain to be elucidated. However, our study identifies PP2Ac methylation as a functional sensor and signaling hub that connects methionine availability to the proliferative capacity of cancer cells. We propose that PP2Ac methylation could serve as a predictive biomarker for methionine dependence, offering a tool to monitor the in vivo effects of methionine-restricted diets, which are being explored as therapeutic strategies in oncology [7,16,91,92,93,94]. Ultimately, deciphering how PP2Ac methylation governs tumor-specific metabolic vulnerabilities may open new avenues for precision therapies that exploit diet–epigenetic–signaling interactions in cancer.

## 5. Conclusions

While the connection between methionine metabolism and cell proliferation is well established, the molecular mechanisms by which methionine availability is sensed and relayed to the cell cycle machinery remain incompletely understood. Although SAMTOR and mTORC1 have been implicated in methionine sensing, our data suggests that this pathway does not underlie methionine dependence in cancer, as mTORC1 signaling remains active even when methionine is depleted and cells are supplemented with homocysteine, conditions under which cancer cells rapidly arrest. This gap in understanding is particularly relevant for cancer biology, as unlike non-tumorigenic cells, many cancer cells are unable to proliferate without exogenous methionine. Elucidating the molecular basis of this methionine addiction is critical for developing targeted therapeutic strategies. Our findings identify PP2A as a functional methionine sensor in mammalian cells and a major driver of methionine addiction in cancer. We show that increasing the demethylated form of PP2Ac is sufficient to convert methionine-independent cells into a methionine-dependent state, demonstrating that PP2A methylation status plays a critical role in linking methionine availability to proliferative capacity. These insights reveal a previously unrecognized mechanism of nutrient sensing that may underlie the selective vulnerability of cancer cells to methionine restriction. Future studies will focus on delineating the downstream signaling pathways regulated by PP2A demethylation and assessing their potential as therapeutic targets in methionine-dependent cancers.

## Figures and Tables

**Figure 1 biomolecules-15-01210-f001:**
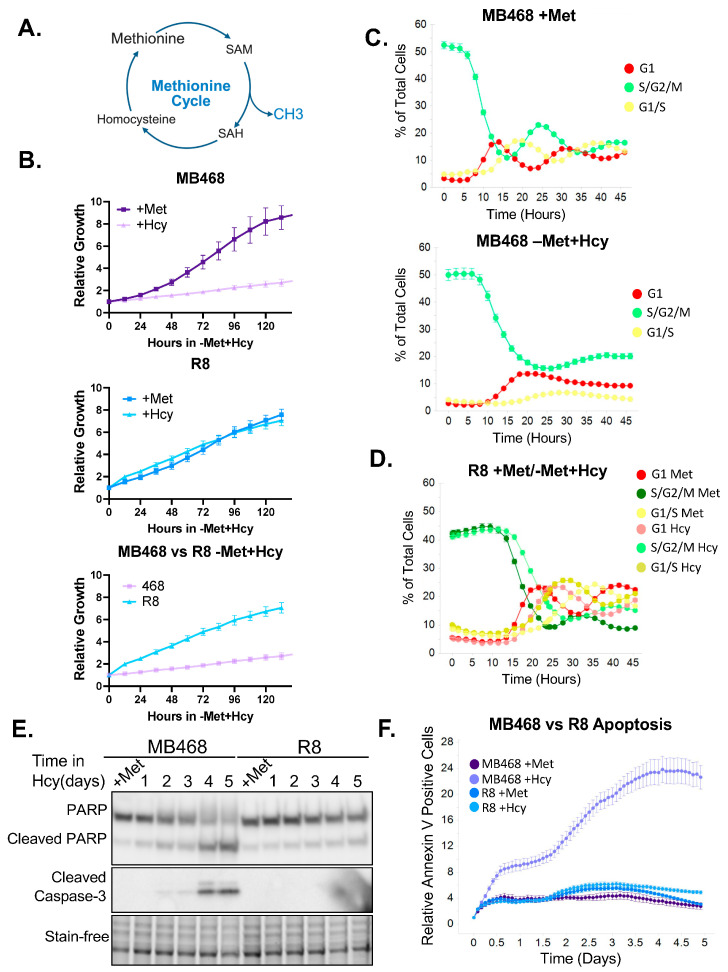
Cancer cells induce cell cycle arrest and cell death when exogenous methionine is depleted. (**A**) The methionine cycle allows for the regeneration of methionine and SAM, which are both essential for cell survival. (**B**) The methionine-sensitive breast cancer cell line MDA-MB468 and the methionine-resistant R8 cell line were seeded in either full media containing 100 µM L-methionine or methionine-depleted media containing 100 µM L-homocysteine instead. Cells were allowed to grow for 6 days. The MB468 cells show a significant growth defect when grown in methionine-depleted (+Hcy) media when compared to the complete (+Met) media. The R8 cells have similar proliferation rates in both media. (**C**) MB468 cells were transfected with Geminin-TagGFP2 and Cdt1-mKate2 to track cell cycle progression using live-cell imaging. Cells were first synchronized in early S-phase using 2.5 mM Thymidine for 24 h. MB468 cells progressed through the cell cycle in complete (+Met) media, but quickly arrested in −Met+Hcy media. Cells did progress from the S/G2/M (green) phases into G1 (red) and then appear to stall, with only a small fraction of the cells entering the G1/S transition (yellow). (**D**) R8 cells were treated under the same conditions as panel C. The cells have normal cell cycle progression in both complete and −Met+Hcy media. (**E**) Both MB468 and R8 cells were grown in −Met+Hcy media, and cells were collected every day for 5 days. Complete media was used as a control. Western blot was used to measure apoptosis by probing for PARP and Caspase-3 cleavage. Low levels of apoptosis were evident as early as day 2, with significant apoptosis after 4 days. R8s exhibit very little PARP cleavage and no Caspase-3 cleavage for the duration of the time course. (**F**) Live-cell imaging using an Annexin-V stain was used to measure apoptosis. The MB468 cells grown in −Met+Hcy media have low levels of apoptosis almost immediately, with the majority of apoptosis induction beginning at day 2 of treatment. MB468s in full media and R8s in both full and methionine-depleted media showed very low levels of apoptosis throughout the experiment. Original figures can be found in Appendix A.

**Figure 2 biomolecules-15-01210-f002:**
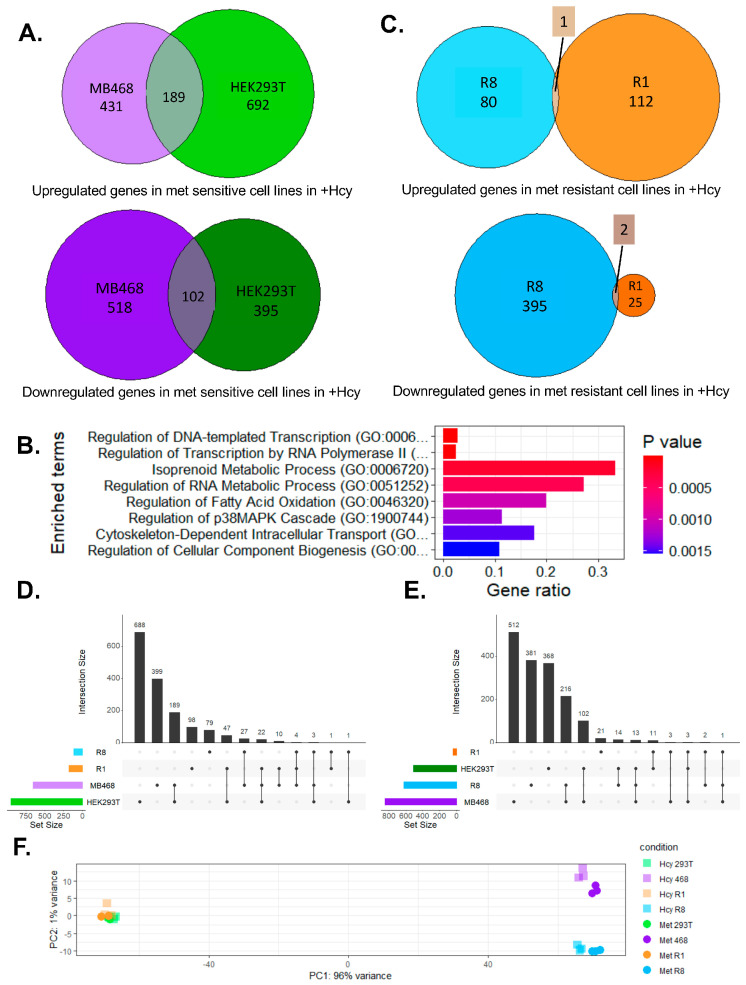
Methionine-independent cells likely have a different mechanism of overcoming Met-dependence. MB468/R8 and HEK293T/Res1 (R1) cell pairs were grown in +Met or −Met/+Hcy media for 12 h and the transcriptional response was analyzed using RNA sequencing. (**A**) Both methionine-dependent cell lines (MB468 and HEK293T) showed a strong transcriptional response in −Met/+Hcy media. (**B**) Gene ontology analysis was performed on genes that are significantly changed in both met-dependent cells, MB468 and HEK293T. (**C**) Comparison between met-independent cell lines R8 and R1 showed only 3 genes overlap in response to methionine restriction (−Met/+Hcy media). (**D**) UpSet plot showing up-regulated and (**E**) down-regulated genes in all cell lines. (**F**) PCA plot demonstrates that the two cell line pairs, 468/R8 vs. 293T/R1 have very different transcriptional responses, with a larger separation between MB468 and R8 cells than the HEK293T/R1 pair.

**Figure 3 biomolecules-15-01210-f003:**
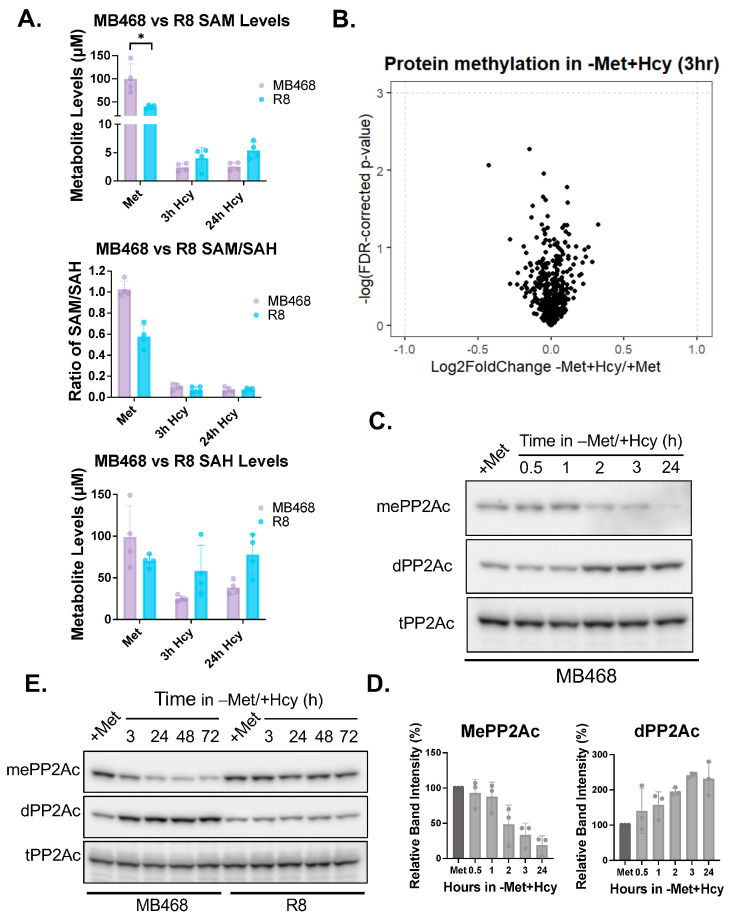
Methyl proteome is largely unchanged, while PP2Ac methylation is specifically sensitive to growth in −Met/+Hcy media. (**A**) MB468 and R8 cells were grown in either +Met or −Met/+Hcy media for 24 h and SAM and SAH levels were measured and compared to standard curves for absolute quantitation. The SAM/SAH ratio was calculated and indicates the methylation potential in the cells. Both met-dependent and met-independent cell lines show a significant decrease in both SAM levels and in methylation potential in response to growth in −Met/+Hcy media. * *p*-value less than 0.05. (**B**) Global protein methylation profiling was done in MB468 cells using liquid chromatography-mass spectrometry. No significant change in the methylproteome was observed after 3 h growth in −Met/+Hcy media. (**C**) Western blot was used to probe for methylated PP2Ac (mePP2Ac), demethylated PP2Ac (dPP2Ac), and total PP2Ac (tPP2Ac). Cells were cultured in −Met/+Hcy media for up to 24 h and harvested at different time points. PP2Ac methylation is lost as early as 2 h of media shift. (**D**) Quantification of E, using *n* = 3 independent experiments. (**E**) MB468 and R8 cells were grown in −Met/+Hcy media for up to 3 days, with samples being collected at different time points. Whole cell lysates were then analyzed for PP2Ac methylation. Original figures can be found in Appendix A.

**Figure 4 biomolecules-15-01210-f004:**
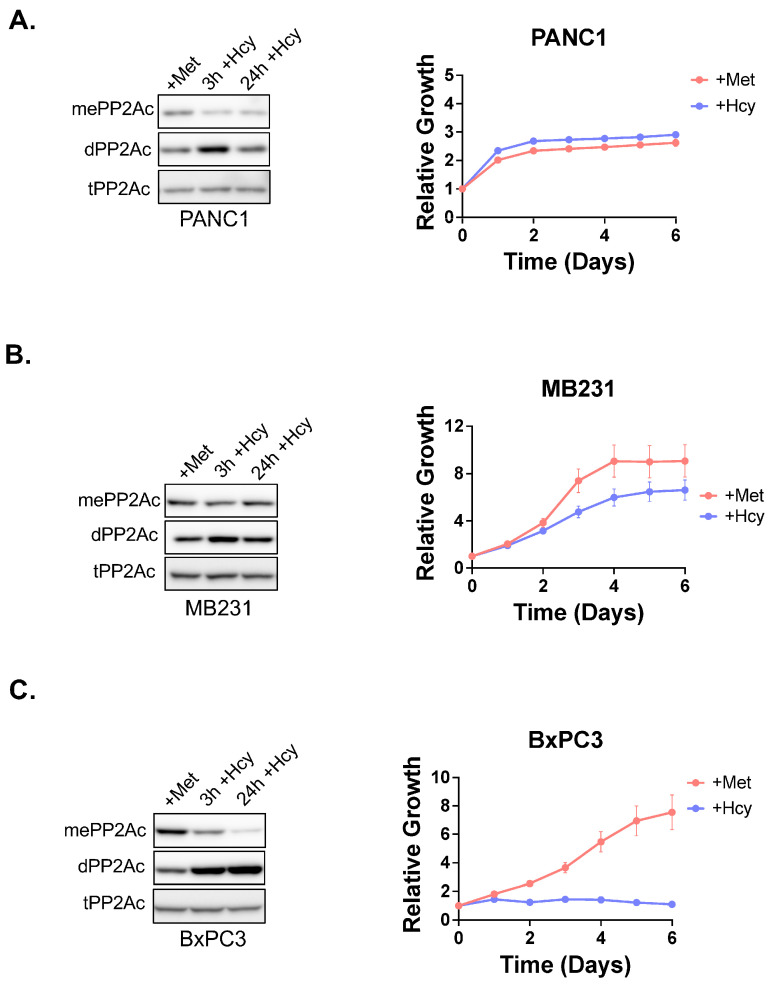
Correlation of cell proliferation in −Met/+Hcy media and PP2Ac methylation across additional cell lines. (**A**) Methionine-independent pancreatic cancer cell line PANC1, (**B**) Methionine-independent breast cancer cell line MDA-MB-231, (**C**) Methionine-dependent pancreatic cancer cell line BxPC3. All cells were all cultured in −Met/+Hcy media and PP2A methylation was analyzed after 3 and 24 h of treatment. Cell proliferation was also measured for 6 days in either +Met or −Met+Hcy media. Original figures can be found in Appendix A.

**Figure 5 biomolecules-15-01210-f005:**
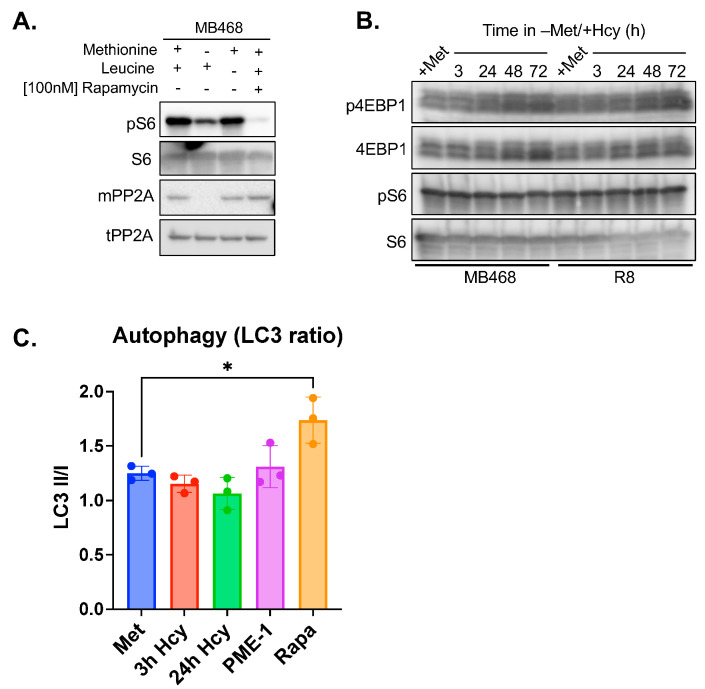
The growth defect of methionine-dependent MB468 cells in −Met/+Hcy media is not mediated by mTor signaling. (**A**) MB468 cells were cultured with either methionine depletion (no homocysteine supplementation), leucine depletion, or 100 nM Rapamycin for 24 h. Whole cell lysates were analyzed for both PP2Ac methylation and S6 phosphorylation. (**B**) MB468 and R8 cells were grown in −Met/+Hcy media for up to 3 days, with samples being collected at different time points. Whole cell lysates were tested for mTORC1 activity by analyzing phosphorylation of mTOR targets. (**C**) LC3II/I ratios indicate that growth in −Met/+Hcy media does not induce autophagy. * *p*-value less than 0.05. Original figures can be found in Appendix A.

**Figure 6 biomolecules-15-01210-f006:**
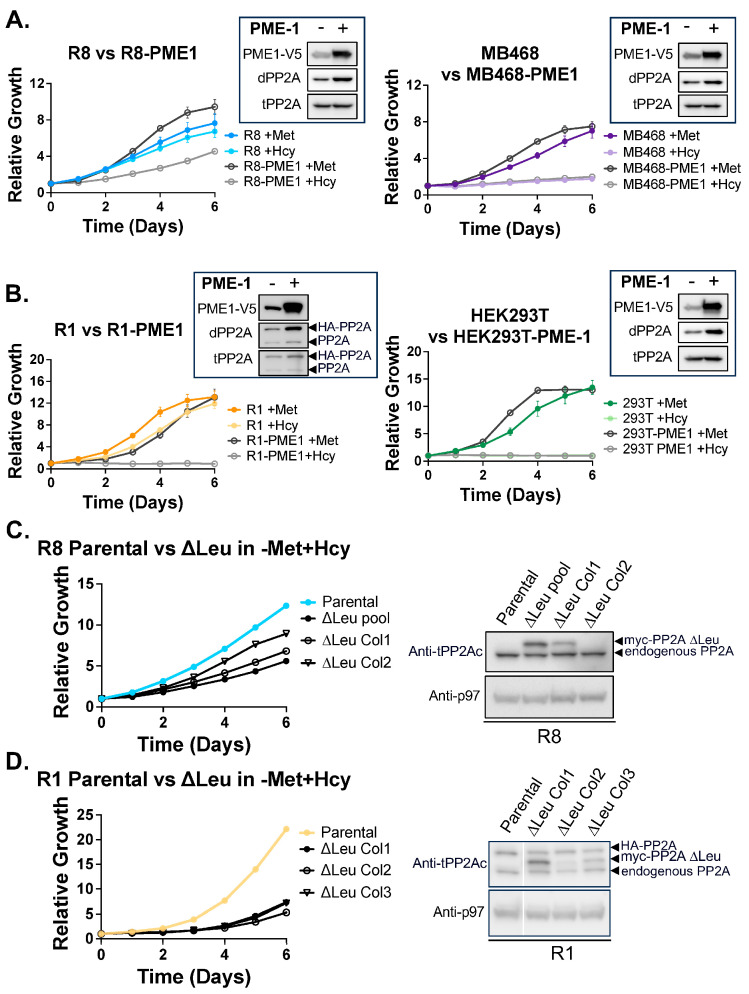
Reducing PP2Ac methylation is sufficient to impair cell proliferation in −Met/+Hcy media. (**A**,**B**) The protein phosphatase methylesterase (PME-1) was overexpressed to decrease PP2Ac methylation in cells. Cell proliferation in +Met or −Met/+Hcy media was compared in both parental and PME-1 overexpressing (PME-1 OE) cells. (**A**) Reducing PP2Ac methylation in R8 cells induced proliferation defects in met-independent R8 cells, while met-dependent MB468 cells were unaffected by PME-1 overexpression. (**B**) Met-independent R1 cells were severely impaired in proliferation in −Met/+Hcy media when PP2Ac methylation was reduced, but HEK293T cells were unaffected by PME-1 overexpression. Note: HA-PP2A- one allele of PP2A was HA-tagged on the endogenous locus. PP2A- endogenous PP2A (**C**) Methionine-independent R8 cells were transduced with myc-PP2Ac ∆Leu309 to mimic demethylation. Growth rates between parental R8, the pool of ∆Leu309 cells, and two individual ∆Leu309 colonies were compared. An increase in demethylated PP2Ac is enough to impair proliferation in −Met/+Hcy media. ∆Leu Col2 had very low expression of the PP2Ac ∆Leu309 allele that was visible only after very long exposure. (**D**) Expression of myc-PP2Ac ∆Leu309 in methionine-independent R1 cells is sufficient to impair proliferation in −Met/+Hcy media. Original figures can be found in Appendix A.

## Data Availability

RNAseq datasets and results are available on NCBI Gene Expression Omnibus (GEO) with accession numbers GSE155955 (MB468, R8) and GSE301557 (HEK293T, R1).

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
