# Peer review of "Carboxy-Methylation of the Catalytic Subunit of Protein Phosphatase 2A (PP2Ac) Integrates Methionine Availability with Methionine Addicted Cancer Cell Proliferation"

_biomolecules, 2025, doi:10.3390/biom15091210_

Round 1
Reviewer 1 Report
Comments and Suggestions for Authors
The authors have identified Protein Phosphatase 2A (PP2Ac) carboxy methylation as a critical vulnerability of methionine - addicted cancer cells when they are restricted of methionine. This finding is a central mechanism of cell -cycle arrest when cancer cells are restricted of methionine.
The authors prove this mechanism by comparing methionine-addicted cancer cells with their methionine - independent revertants which do not have this vulnerability of reduced methylation of PP2Ac when they are restricted of methionine. The experiments are exquisitely carried out. Minor ( but helpful) points: 1) The title should be modified to state ".... methionine- addicted cancer cells..." and PP2Ac should be spelled out. 2) The original study that showed a sharp drop in SAM upon methionine restriction of cancer cells was published in 1982 ( Proc Natl Acad Sci USA 79(14): 4248-4251, 1982). 3) Methionine addiction is very tightly linked to malignancy (iScience 25(4): 104162,2022 ). 4) Papers stating cancer cells have reduced or defective methionine synthase (MS) are probably wrong because without a functional MS folate is trapped as 5-methyl tetrahydrofolate (5-MethylTHF) and cancer cells would not be able grow in vivo where 5-MethylTHF is the main circulating folate (Nat Metab 3(11): 1500-1511, 2021; Nat Metab 3(11): 1512-1520, 2021). 5) Cancer cells that grow in MET-HCY+ medium are still more sensitive to methioninase than normal cells and therefore also methionine addicted (Anticancer Res 30(4): 1041-1046, 2010). 6) It was shown in 1980 that cancer cells which are transformed by T-antigen arrest in late-S/G2 upon methionine restriction (Proc Natl Sci USA 77(12): 7306-7310, 1980).Author Response
Please see the attachment.

Reviewer 2 Report
Comments and Suggestions for Authors
In this work by Andronicos et al, the authors sought to understand how methionine metabolism impacts cell cycle progression, by focusing on the established growth defects of many cancer cells in the absence of methionine. The authors conducted a comprehensive study incorporating cell growth assays, cell cycle analysis, transcriptomics, metabolomics, and proteomics to study the role of methionine metabolism in the cell cycle. In this work, the dependency of the cell cycle on methionine is revealed, and PP2Ac emerges as a critical regulator of methionine dependency in cancer. This is overall an impactful study that significantly enhances our understanding of cancer cell metabolism and biology. I have a few concerns and suggestions, that if implemented, would enhance the manuscript:
- In the discussion, the authors claim "Importantly, our results demonstrate that loss of PP2Ac methylation is necessary for cell cycle arrest and cell death in methionine-dependent cancer cells specifically when grown in -Met+Hcy medium." However, the effect of PP2Ac methylation on cell cycle is not explicitly demonstrated, only impact on cell proliferation. Analysis of cell cycle would be useful and strengthen the manuscript, especially if the PP2Ac ∆Leu cells happen to stall in the same place in the cell cycle as cells cultured -Met.
- The results in Figure S11 are significant, and demonstrate that PP2Ac methylation is relevant across multiple cell lines. The authors should introduce this data in the results section, rather than the discussion.
- In Figure 5C, does Col2 have myc-PP2Ac ∆Leu expression? The western blot is not convincing. Does over expression of WT (myc-PP2Ac) have any impact on cell function, or is this the "parental" cell line referenced?
- The authors should carefully check that all supplementary data is correctly referenced. One or two supplemental figures were referenced in the text improperly (this is a minor concern).
Reviewer 3 Report
Comments and Suggestions for Authors
Cancer cells are dependent upon the availability of exogenous methionine for growth, but the metabolic and the molecular background of this phenomenon is not clear yet. In this manuscript, Andronicos et al. investigated how methionine metabolism affects cell cycle progression by using methionine dependent and independent cell lines. They provide impressive sets of data on the research aims including novel findings on this topic suggesting that the methionine dependent proliferation/growth effects are due to changing in the methylation level of the catalytic subunit of protein phosphatase 2A (PP2Ac) at the carboxy-terminal Leu. Although, it was shown earlier that methylation of PP2Ac impacts cell cycle, the present findings of the authors have novelty in context of methionine dependent cancer cell proliferation. Convincing data are presented for the role of PP2A methylation using methylation specific antibody, methylesterase overexpression as well as deletion of the methylation target Leu. The experimental results are well documented, even though; one may miss a few additional experiments which could have been carried out with the samples used in this study
Critical notes:
1. It is claimed that the phosphatase activity did not change with varying methylation states of PP2Ac. However, in the methods activity measurement is described only for HA-tagged PP2Ac. What is the effect of methylation on the activity of endogenous PP2A.
2. The p-nitrophenyl phosphate is rarely used to assay protein phosphatases. Using a phospho-protein or -peptide is a better approach to determine cellular phosphatase activity.
3. PP2Ac methylation primarily affects association of the core dimer (C and A subunit) with the B subunits. It would have been a good supplement to look for B subunit (preferably Bα) changes in the methylated and demethylated PP2Ac samples.
4. In the supplementary on Fig. S9 the Western blots signals for the different PP2A forms are quite weak and faint, not suitable to draw any consistent conclusion. Redo the blots.
Round 2
Reviewer 1 Report
Comments and Suggestions for Authors
The manuscript is in condition for acceptance.
Reviewer 2 Report
Comments and Suggestions for Authors
The authors addressed my concerns and the manuscript is suitable for acceptance.